# Effect of Modified Natural Rubber on the Mechanical and Thermal Properties of Poly(Lactic Acid) and Its Composites with Nanoparticles from Biowaste

**DOI:** 10.3390/polym16060812

**Published:** 2024-03-14

**Authors:** Preeyaporn Injorhor, Supharat Inphonlek, Yupaporn Ruksakulpiwat, Chaiwat Ruksakulpiwat

**Affiliations:** 1School of Polymer Engineering, Institute of Engineering, Suranaree University of Technology, Nakhon Ratchasima 30000, Thailand; preeyaporn_in@sut.ac.th (P.I.); supharat.inph@gmail.com (S.I.); 2Research Center for Biocomposite Materials for Medical Industry and Agricultural and Food Industry, Nakhon Ratchasima 30000, Thailand

**Keywords:** poly(lactic acid), modified natural rubber, toughening agent, hydroxyapatite

## Abstract

The brittle behavior of poly(lactic acid) (PLA) and PLA composites with inorganic filler limits their applications; the addition of a toughening agent, such as a rubbery phase, was selected to transform the brittle to ductile behavior for versatility in various applications. This work aims to study the properties of PLA and PLA composite with filled nanosized hydroxyapatite (nHA) after adding modified natural rubber (MoNR), which acts as a toughening agent. MoNR refers to poly(acrylic acid-co-acrylamide)-grafted deproteinized natural rubber. nHA was prepared from fish scales. Its characteristics were investigated and was confirmed to be comparable to those of commercial grade. PLA-MoNR at various MoNR contents and PLA/nHA composites with/without MoNR were prepared by melt mixing. Their morphology, mechanical, and thermal properties were observed and investigated. Samples with MoNR added showed the dispersion of spherical particles, indicating incompatibility. However, the mechanical properties of PLA-MoNR, which had MoNR added at 10 phr, showed toughening behavior (increased impact strength by more than two times compared to that of neat PLA). The PLA/nHA composite with MoNR showed the same result. The addition of MoNR in the composite increased its impact strength by 1.27 times compared to the composite without MoNR. MoNR can be a stress concentrator, resulting in toughened PLA and PLA/nHA composite.

## 1. Introduction

Due to the concern regarding sustainable development, sustainable and biobased polymers, derived from renewable feedstocks, are receiving attention worldwide [1,2]. Furthermore, they contribute to the biobased economy, which is an economic, social, and environmental model aiming to build a sustainable society [3]. Sustainable and biobased polymers, such as natural rubber (NR), chitin, cellulose, poly(lactic acid) (PLA), poly(hydroxyalkanoate)s (PHAs), poly(ε-caprolactone) (PCL), etc., are part of this growing focus on environmentally friendly materials [4].

PLA is an aliphatic polyester derived through the fermentation of renewable resources such as sugarcane, corn, rice, etc. [5]. Its outstanding properties include being highly rigid, biodegradable, biocompatible, renewable, and recyclable [2,6,7,8]. Consequently, it is an interesting biodegradable polymer with major applications in packaging, automotive, and biomedical fields [9,10,11,12,13]. Unfortunately, PLA is too brittle and has low impact resistance at room temperature due to its quite low glass-transition temperature, limiting its utilization. Many researchers have attempted to improve the toughness of PLA by incorporating a rubbery phase into PLA, a process known as rubber toughening [12,14,15,16]. The dispersion of rubber micro or nanoparticles in a brittle polymer can provide energy absorption through the initiation of local yielding [17]. This polymer modification is expected to improve toughness and elongation at break while retaining strength [17].

NR is one of Thailand’s mass-produced feedstocks. NR, derived from *Hevea brasiliensis*, attracts the interest of many researchers due to its outstanding characteristics, including flexibility, high strength, biocompatibility, and biodegradability [18]. It is a simple rubber selected for use as a toughening agent because of its unique properties like high elasticity and modulus. However, it must be modified because it is a non-polar rubber that is immiscible with PLA [19,20,21,22]. Previously, NR was modified by grafting with polar monomers such as poly(vinyl acetate), PVAc [23]; poly(acrylic acid), PAA [20]; and poly(butyl acrylate), PBA [19] to be used as a toughening agent for PLA-NR blends. However, a copolymer was not selected to graft onto NR for use as a toughening agent in PLA and PLA composites. This work studies the effect of NR-grafted copolymer on the mechanical and thermal properties of PLA and PLA composites. Poly(acrylic acid-co-acrylamide)-grafted deproteinized natural rubber was selected for use as a toughening agent, and it was prepared according to a previous work [24]. Due to the polarity of poly(acrylic acid-co-acrylamide)-grafted deproteinized natural rubber, it has been incorporated with other materials and successfully utilized in the functions of controlled-release fertilizer and dye removal [25,26]. Therefore, it is possible that it could serve as a toughening agent.

Additionally, a bio-filler in PLA composites, such as hydroxyapatite (HA) from fish scale waste, was prepared by alkali heat treatment. The preparation was improved by using an autoclave to reduce preparation time. It is expected that the size of the obtained HA is still in the range of nanoscale, even though the preparation time was reduced. Previously, HA was used as a filler in PLA and PLA-PHA composites for medical application expectation due to its osteoconductive properties. Regarding to mechanical properties, it can improve tensile strength and Young’s modulus. On the contrary, the elongation at break was reduced by adding HA [27]. Modified natural rubber (MoNR), refer to poly(acrylic acid-co-acrylamide)-grafted deproteinized natural rubber was expected to improve elongation at break and toughness of PLA and PLA composite in this study.

Ultimately, the improvement of PLA properties leads to the development of materials from sustainable resources, which can serve as substitutes for materials derived from non-renewable resources.

## 2. Materials and Methods

### 2.1. Materials

Poly(lactic acid) (PLA, Ingeo™ Biopolymer 4043D, General Purpose Grade) was supplied by Nature Works LLC (Minnetonka, MN, USA). Poly(acrylic acid-co-acrylamide)-grafted deproteinized natural rubber had a grafting efficiency and grafting percentage of 20.8% and 2.1%, respectively. In this study, it was called MoNR, which was used as a toughening agent. It was prepared in-house, as described in [24]. Nanosized hydroxyapatite (nHA) was prepared from dried seabass scales purchased from a local store in Chonburi, Thailand. For comparison, commercial HA with a particle size of 5 µm was purchased from Sigma-Aldrich (St. Louis, MO, USA). Hydrochloric acid (HCl) 37% RPE and sodium hydroxide (NaOH) 99% RPE-ACS were purchased from Carlo Erba (Milano, Italy).

### 2.2. Preparation of nHA from Seabass Scales

nHA was prepared from dried seabass scales. The method of the preparation was adapted from a previous work [28]. The dried scales were treated with 0.1 M HCl at room temperature for 30 min. After that, they were washed with deionized water until the pH was neutral before being oven-dried at 80 °C. Then, the product from the previous step was alkali heat-treated with 10% w/v NaOH solution, heated at 121 °C, 15 psi for 1 h in an autoclave (SX-700, TOMY, Fremont, CA, USA). Then, the scales became small particles and precipitated at the bottom of the bottle. The only supernatant was drained and the precipitate was washed until the pH was neutral with deionized water. Then, the precipitate was frozen at −80 °C and then lyophilized by a freeze dryer (Alpha 2–4 LSC, Chirist, Osterode am Harz, Germany) for 1 day. The final product become the nHA powder with 39.38% yield of preparation.

### 2.3. Preparation of PLA-MoNR and PLA/nHA Composites with/without MoNR

PLA-MoNR with various MoNR contents (2.5, 5, and 10 phr) and the PLA/nHA composites with/without MoNR were prepared. Their compositions are shown in Table 1. All samples were prepared by an internal melt mixer (Rheomix OS, HAAKE, Thermo Scientific, Waltham, MA, USA) at a temperature of 175 °C for 10 min with a rotor speed of 60 rpm. Then the samples were cut to reduce the size into granules. To obtain a dumbbell specimen (Type V ASTM D638-14 [29]) for tensile testing and a bar specimen for Izod impact testing, both samples were prepared by compression molding at 180 °C for 5 min. Before compressed, the mixture was preheated for 10 min at the same temperature and pumped 10 times.

### 2.4. Characterization

The crystallographic structures of nHA and commercial HA were characterized using an X-ray diffractometer (XRD, D2 PHASER, Bruker, Billerica, MA, USA) with Cu Kα radiation source. This was operated at 30 kV and current 10 mA. Bragg’s angle of diffraction (2θ) was varied from 10 to 60 at a scan rate of 2°/min, with a step size of 0.02°.

The nHA and commercial HA were determined the chemical functional groups by using a Fourier transform infrared spectrometer (FTIR, Tensor 27, Bruker, Billerica, MA, USA). The measurements were taken with 64 scans with a resolution of 4 cm^−1^, in the frequency range of 400 cm^−1^ to 4000 cm^−1^.

The microstructure and size of nHA was observed using a field emission scanning electron microscope (FE-SEM, Carl Zeiss Auriga, Oberkochen, Germany) at 3 kV of acceleration voltage. The samples were sputter-coated with gold for 3 min at 10 mA before the testing was carried out.

The energy-dispersive X-ray spectrum and chemical elemental composition of nHA were investigated using an energy-dispersive X-ray spectroscope (EDS, EDAX Genesis 2000, AMETEX, Berwyn, PA, USA) in a scanning electron microscope (SEM, JSM-6010LV, JOEL, Akishima, Tokyo, Japan).

The morphologies of the neat PLA, PLA-MoNR, and PLA/nHA composites with/without MoNR from the tensile fracture and nitrogen-frozen fracture were observed by using SEM (SEM, JEOL, JSM-6010LV, Tokyo, Japan). All samples were sputter-coated with gold for 3 min at 10 mA before the procedure was carried out.

The mechanical properties of the neat PLA, PLA-MoNR, and PLA/nHA composites with/without MoNR were measured by tensile testing and impact testing. For tensile testing, the properties such as tensile strength, Young’s modulus, and elongation at break were investigated according to ASTM D638-14 using a universal testing machine (UTM, INSTRON/5565, Norwood, MA, USA) with a load cell of 5 kN and a crosshead speed of 10 mm/min at room temperature. For impact testing, notched Izod impact testing was performed according to ASTM D256-10 [30] using a pendulum impact testing machine (INSTRON, Norwood, MA, USA).

Thermal properties of the neat PLA, PLA-MoNR, and PLA/nHA composites with/without MoNR were investigated using a differential scanning calorimeter (DSC, DSC 3+ STARe System, Mettler Toledo, Schwerzenbach, Switzerland). The samples were heated from −80 to 200 °C with a heating rate of 20 °C/min, under nitrogen at a flow rate of 50 mL/min, followed by a cooling process down to −80 °C and second heating with the same procedure. The DSC thermograms provide the thermal properties such as enthalpy of melting (ΔHm), enthalpy of cold crystallization (ΔHcc), glass transition temperature (Tg), cold crystallization temperature (Tcc), and melting temperature (Tm). The degree of crystallinity (Xc) was calculated according to Equation (1) [27]:Χc (%) = [(ΔHm − ΔHcc)/(ΔHm0 × w)] ×100(1)
where ΔHm0 is the heat of melting of purely crystalline PLA (93 J·g^−1^), and w is the weight fraction of PLA in each sample.

The thermal stability of the neat PLA, PLA-MoNR, and PLA/nHA composites with/without MoNR was characterized using a thermal gravimetric analyzer (TGA, TGA/DSC1, Mettler Toledo, Schwerzenbach, Switzerland) under a nitrogen atmosphere from 50 °C to 500 °C with a flow rate of 50 mL/min and a heating rate of 10 °C/min.

## 3. Results and Discussion

### 3.1. Characteristics of nHA

#### 3.1.1. XRD Analysis

Figure 1 illustrates the XRD patterns of nHA powder and commercial HA, which were matched with each other and with the XRD standard data of the characteristic spectrum of hydroxyapatite (JCPD card-PDF 9-432) [31]. This confirms that the nHA prepared from seabass scales is mostly composed of crystalline HA with single-phase HA. Additionally, nHA was compared to commercial HA. It was confirmed to have a similar crystal structure. This results are consistent with that of a previous work [28], demonstrating that this method can reduce preparation time by using an autoclave in the alkali heat treatment step. In the work by Amanda et al. [32], a rapid method for recycling cigarette butts into cellulose was presented. They concluded that an autoclave can reduce the deacetylation time required to turn cigarette butts into cellulose.

#### 3.1.2. FTIR Analysis

The chemical functional groups of nHA and commercial HA were determined using FTIR to confirm the similar chemical functional groups. The FTIR spectra of nHA and commercial HA are presented in Figure 2. In the case of commercial HA, the characteristic hydroxyl group (OH) peaks in the HA structure appear at 3467 cm^−1^ and 634 cm^−1^ [33], while nHA exhibits only a tiny peak at 669 cm^−1^. The observed shift of the peak to a higher wavenumber is attributed to the lower OH concentration in the nHA structure [34]. The OH stretching of adsorbed water is indicated by the broad peak around 3400–3500 cm^−1^, with an additional peak at 1627 cm^−1^ corresponding to the bending of the OH groups in water molecules (H_2_O) [31]. Several peaks are observed in the nHA spectrum, including a tiny peak at 1548 cm^−1^, peaks at 1454, 1416 cm^−1^, and a tiny sharp peak at 875 cm^−1^. These are attributed to carbonate groups (${CO}_{3}^{2-})$ substituting phosphate groups $\left({PO}_{4}^{3-}\right)$ in the nHA structure [35]. Additionally, ${CO}_{3}^{2-}$ are also present in commercial HA, likely due to the incorporation of carbon dioxide (${CO}_{2})$ during the processing of commercial HA [36], a phenomenon commonly found in commercial HA [37,38]. However, both nHA and commercial HA exhibit peaks related to ${PO}_{4}^{3-}$ in the range of 1124–1000 cm^−1^, with small peaks at 962, 601, and 567 cm^−1^ [28].

#### 3.1.3. Microstructure Analysis

The microstructure of nHA is depicted in Figure 3. Nanosized bio-particles, identified as nHA, were successfully prepared from seabass scales. The nHA particles exhibit an irregular shape, with a size of approximately 100 nm as compared to the scale bar. However, some areas of the nHA microstructure also show agglomerated particles.

#### 3.1.4. Chemical Elemental Composition Analysis

The chemical elemental composition of nHA was examined by EDS to determine the atomic Ca/P ratio. The resulting spectrum and chemical elemental composition are presented in Figure 4 and Table 2. The atomic Ca/P ratio is 1.72, indicating non-stoichiometric hydroxyapatite or calcium-rich nHA [39,40]. However, the use of an autoclave for alkali heat treatment proved to be one of the successful methods for eliminating collagen and other organic constituents from raw fish scales. Importantly, this process achieved the removal of these components without adversely affecting the molecular structure of the resulting HA.

### 3.2. Characterization of Neat PLA, PLA-MoNR at Various MoNR Contents, and PLA/nHA Composites with/without MoNR

#### 3.2.1. Mechanical Properties

The mechanical properties were investigated through tensile and Izod impact testing. Figure 5 shows tensile stress-strain curves of all the samples and Table 3 presents Young’s modulus, tensile strength, elongation at break, and impact strength for neat PLA, PLA-MoNR, and PLA/nHA composites with/without MoNR. In the case of PLA-MoNR, Young’s modulus decreased with increasing MoNR content due to immiscibility and poor interfacial adhesion between PLA and MoNR [16,41]. At 5 phr of MoNR, the highest elongation at break was observed (75.5 ± 13.0%). According to Pongtanayut et al. [16], the softness and flexibility of rubber can increase the blend’s elongation, but the dispersion and size distribution of the rubber phase are crucial factors. Better dispersion and size distribution of the rubber phase in the PLA matrix resulted in more elongation. Therefore, PLA with 5 phr of MoNR exhibited the best dispersion, leading to the highest elongation. For the PLA/nHA composite with MoNR, MoNR also reduced tensile strength and modulus but increased elongation for the same reasons. Regarding impact strength, PLA with 10 phr of MoNR exhibited the highest impact strength because MoNR acted as a stress concentrator in PLA-MoNR and PLA/nHA composite with MoNR. The higher content of the rubber phase enhanced fracture energy absorption [19,42]. As seen in Table 3, impact strength increased with increasing MoNR content. This result aligns with that of the PLA/nHA composite with MoNR, where MoNR also enhances impact strength. Although the addition of MoNR may improve toughness, it is unable to preserve other mechanical properties such as tensile strength and Young’s modulus. So, the utilization should be considered all-around properties.

#### 3.2.2. Morphology

Figure 6 displays SEM micrographs of the tensile and freeze-fractured surfaces of neat PLA, PLA-2.5MoNR, PLA-5MoNR, PLA-10MoNR, PLA-5MoNR/2.5nHA composite, and PLA/2.5nHA composite. Both surfaces of neat PLA appear smooth and homogeneous, indicative of a brittle fracture. In contrast, PLA-MoNR exhibits spherical particles dispersed in the PLA matrix, showcasing a ductile deformation behavior, particularly in PLA-5MoNR and PLA-10MoNR. The spherical particles are identified as MoNR particles, which are immiscible with PLA. The number of MoNR particles in the PLA matrix varies with the added MoNR content. Notably, the freeze-fractured surfaces on the left side reveal a higher number of particles at 2.5 and 10 phr of MoNR compared to 5 phr. This suggests that 5 phr may be an optimal content compatible with PLA, leading to some MoNR particles and PLA becoming compatible, resulting in the disappearance of spherical particles [19]. Additionally, the morphology of the PLA/nHA composite without MoNR exhibits a homogeneous surface without spherical particles and demonstrates a brittle fracture. However, it possesses surface layers unlike neat PLA due to the addition of nHA. Furthermore, the PLA/nHA composite with MoNR shows a ductile deformation behavior. In summary, the overall morphology results indicate that the addition of MoNR can transform brittle PLA into a ductile PLA, including the PLA/nHA composite.

#### 3.2.3. Thermal Properties

The thermal behaviors of neat PLA, PLA-MoNR, and PLA/nHA composites with/without MoNR were analyzed based on the values of glass transition temperature (Tg), enthalpy of cold crystallization (∆Hcc), enthalpy of melting (∆Hm), melting temperature (Tm), and crystallinity (Xc), which were measured by DSC. The ∆Hcc, ∆Hm, and Xc values along with the thermograms during the first and second heating scan of neat PLA, PLA-MoNR, and PLA/nHA composites with/without MoNR are presented in Table 4 and Figure 7, Figure 8 and Figure 9, respectively.

During the first heating scan, the crystallinity of each sample does not correlate with their toughness, especially when considering impact strength. It may be concluded that the toughness of the samples was enhanced by the effective dispersion of MoNR particles. However, these results indicate that the addition of MoNR and nHA, either individually or in combination, does not significantly increase the crystallinity of PLA. For the second heating scan, the samples had the heat profile eliminated from the first heating scan. The results showed the same thermal behavior, indicating that the addition of MoNR and nHA does not affect the crystallinity of PLA.

The thermal properties of neat PLA, PLA-MoNR, and PLA/nHA composites with/without MoNR obtained from the second heating scan are listed in Table 5. The second heating scan DSC thermograms of neat PLA and PLA-MoNR at various MoNR contents are depicted in Figure 8. An evident endothermic peak was observed at approximately −60 °C, indicating the glass transition temperature (Tg) of MoNR in PLA-MoNR at 5 and 10 phr of MoNR. The Tg of PLA was not altered by the addition of MoNR, likely due to poor compatibility between PLA and MoNR [18]. However, the slight variations in the Tcc and Tm of PLA could be attributed to the disruption caused by MoNR and nHA particles during cold crystallization, as well as the potential reorganization within the crystalline phase upon heating [43]. The addition of MoNR did not have a substantial impact on the Tg of PLA. It can be concluded that the immiscibility of both binary and ternary compositions resulted in an unchanged thermal behavior. Considering the PLA composites, as illustrated in Figure 9, small changes in Tg, Tcc, and Tm were detected. All values decreased with the addition of nHA. This decrease in Tcc can be elucidated by the capacity of nHA particles to serve as efficient nucleation sites for PLA crystallization. The decrease in Tg and Tm can be linked to an increase in the free volume and chain mobility of the PLA composite with nHA added, which results from the loss of PLA molecular weight during processing [44]. Additionally, the addition of MoNR in the PLA composite showed a slight decrease of Tg and Tm, but Tcc did not change.

The thermal stability of neat PLA, PLA-MoNR, and PLA composites was investigated using the TGA technique, providing the onset decomposition temperature ${\;(T}_{Onset})$ and the temperature of maximum degradation rate ${(T}_{Max})$ for PLA-MoNR and PLA composites. These values are listed in Table 6; the one-stage weight loss thermal decomposition is illustrated in Figure 10 and Figure 11. All samples exhibited one-stage weight loss during thermal decomposition. However, both $T_{Onset}$ and ${\;T}_{Max}$ of PLA-MoNR increased with rising MoNR content, indicating that MoNR enhances the thermal stability of the PLA-MoNR compared to neat PLA. This improvement can be attributed to the interaction between the functional groups of PLA and MoNR, leading to thermal resistance and consequently enhancing the thermal stability, consistent with the findings of Udomkitpanya and Srikulkit [20]. Moreover, Juntuek et al. [45] also noted that the thermal stability of PLA increased with the addition of NR, attributed to good interaction between PLA and NR, along with the effective dispersion of NR in the PLA matrix. Similar results were observed for the PLA/nHA composite samples, showing improved thermal stability compared to neat PLA, even without MoNR. However, when PLA-5MoNR and PLA-5MoNR/2.5nHA were compared, a different outcome was observed. PLA-5MoNR/2.5nHA exhibited earlier degradation than PLA-5MoNR, possibly due to a decrease in molecular weight during the melt mixing process, leading to a reduced $\;T_{Onset}$. This result aligns with the findings reported by Bijarimi et al. [44].

## 4. Conclusions

The preparation of nHA from fish scales was successfully conducted through alkali heat treatment using an autoclave to reduce preparation time. The obtained nHA was similar to commercial HA in terms of the crystal structure and chemical functional groups. It was used as a filler in PLA/nHA composites with/without MoNR to study the effect of MoNR on the properties of the PLA/nHA composite. PLA-MoNR and PLA/nHA composites with/without MoNR were prepared using an internal melt mixer. MoNR can act as a toughening agent in PLA-MoNR, especially with the addition of MoNR at 5 and 10 phr. The PLA-MoNR with 5 phr of MoNR is the optimum content that provided the best elongation, while the PLA-MoNR with 10 phr of MoNR shows the highest impact strength. The addition of MoNR does not affect thermal behavior obviously, indicating that PLA and MoNR are immiscible. However, the addition of MoNR led to an improvement in elongation, impact strength, and thermal stability. Regarding PLA/nHA composites with/without MoNR, MoNR also acts as a toughening agent in the PLA/nHA composite with MoNR. However, the combination of MoNR and nHA may reduce thermal stability of the PLA/nHA composite with MoNR.

## Figures and Tables

**Figure 1 polymers-16-00812-f001:**
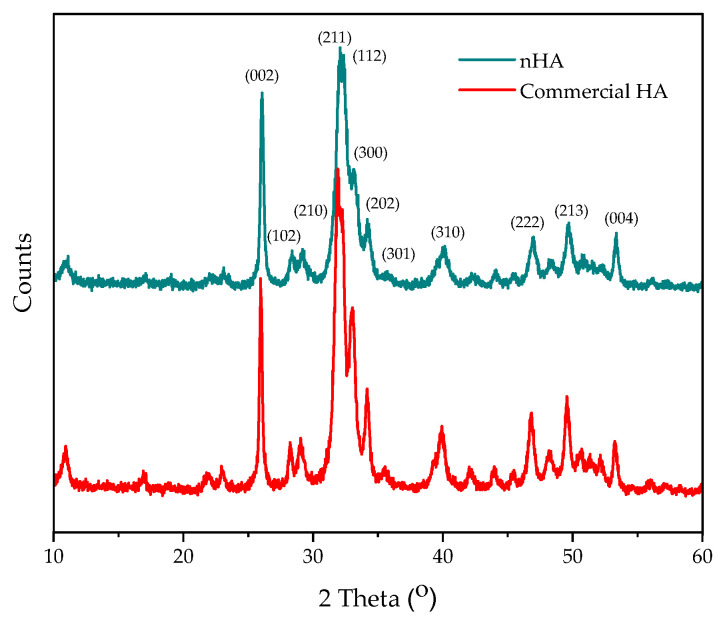
XRD patterns of nHA powder and commercial HA.

**Figure 2 polymers-16-00812-f002:**
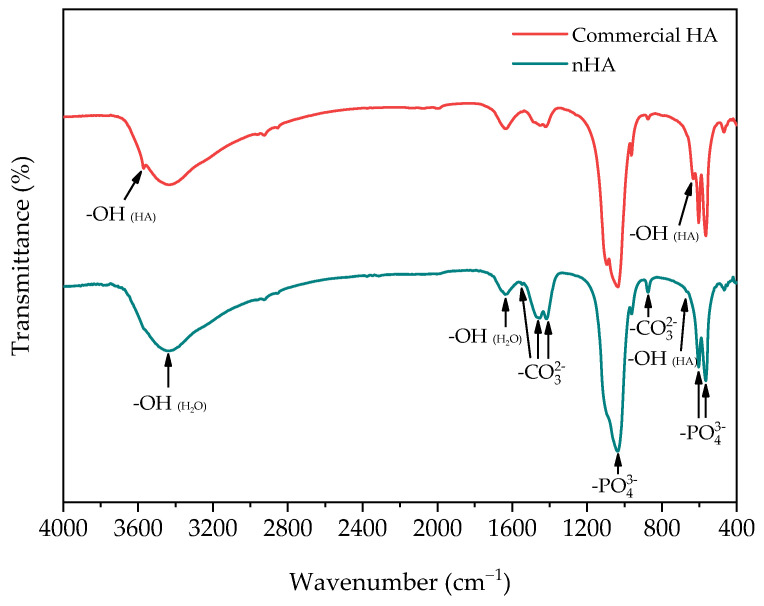
FTIR spectra of nHA powder and commercial HA.

**Figure 3 polymers-16-00812-f003:**
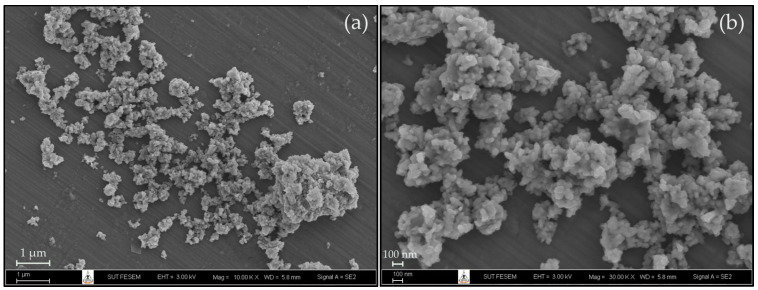
FE-SEM images of nHA powder at (**a**) 10,000×, (**b**) 30,000× magnification.

**Figure 4 polymers-16-00812-f004:**
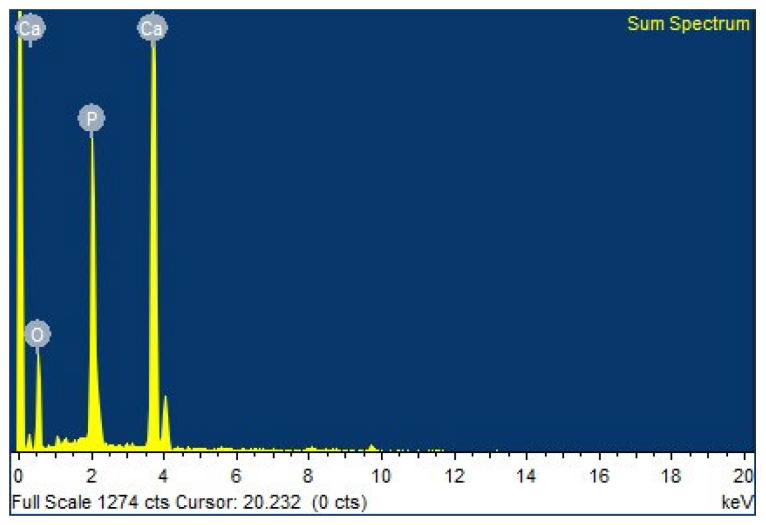
EDS spectrum of nHA powder.

**Figure 5 polymers-16-00812-f005:**
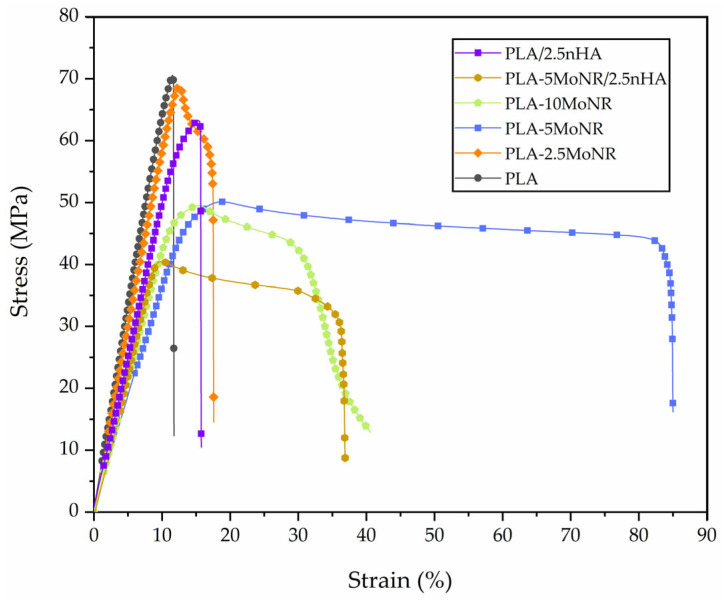
Tensile stress-strain curves of neat PLA, PLA-MoNR at various MoNR contents, and PLA/nHA composites with/without MoNR.

**Figure 6 polymers-16-00812-f006:**
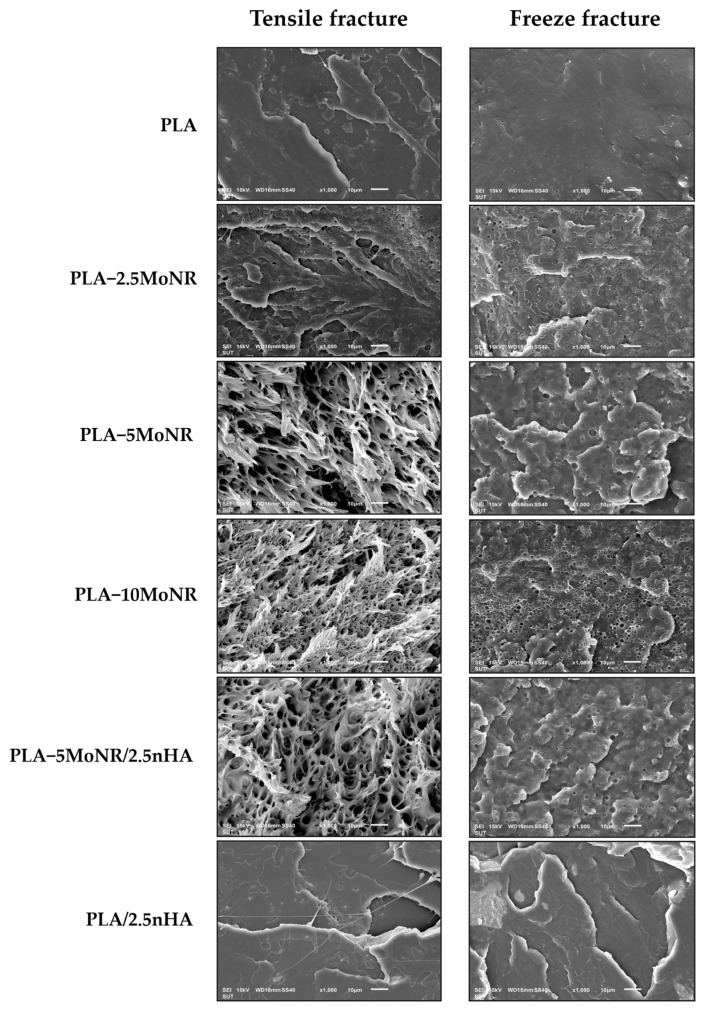
SEM micrographs showing tensile fractured (right) and freeze fractured (left) of neat PLA, PLA-MoNR at various MoNR contents, and PLA/nHA composites with/without MoNR.

**Figure 7 polymers-16-00812-f007:**
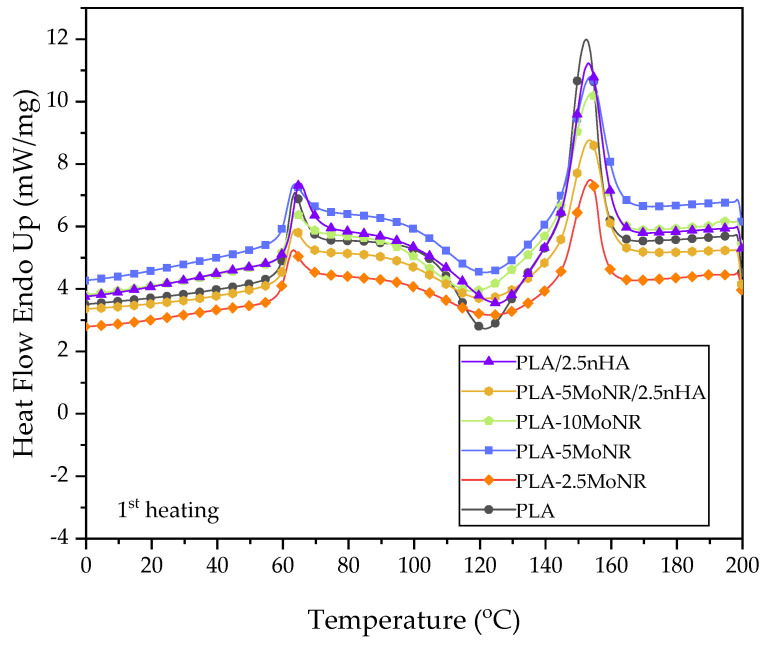
First heating scan DSC curves of neat PLA, PLA-MoNR at various MoNR contents, and PLA/nHA composites with/without MoNR.

**Figure 8 polymers-16-00812-f008:**
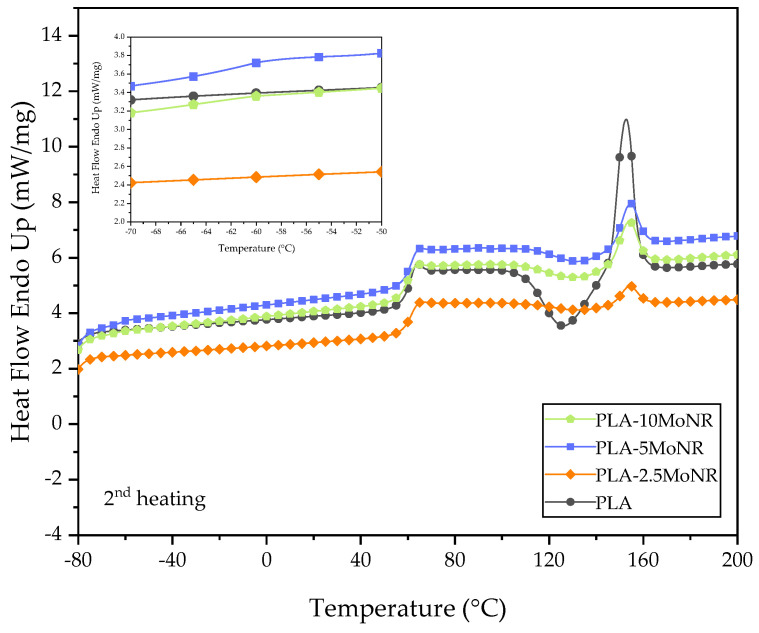
Second heating scan DSC curves of neat PLA and PLA-MoNR at various MoNR contents.

**Figure 9 polymers-16-00812-f009:**
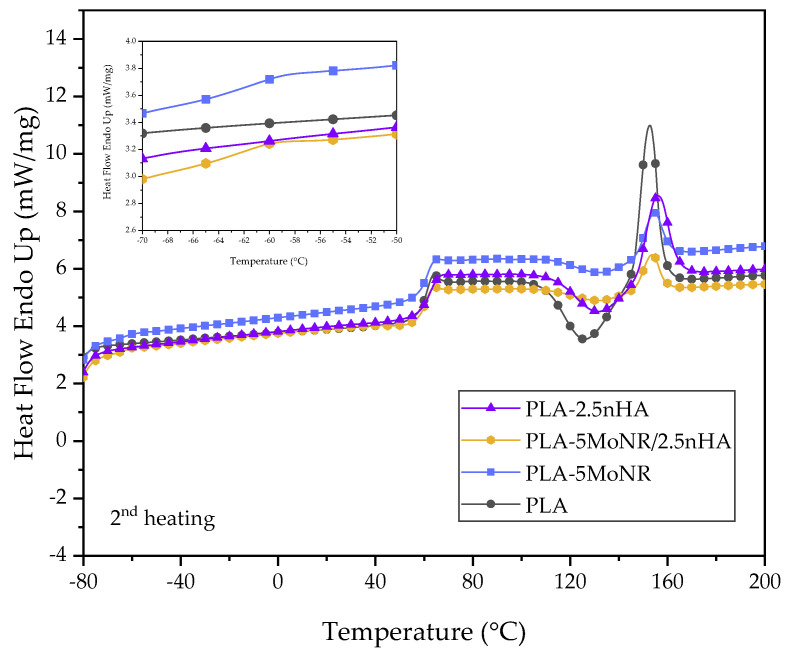
Second heating scan DSC curves of neat PLA, PLA-5MoNR, and PLA/nHA composites with/without MoNR.

**Figure 10 polymers-16-00812-f010:**
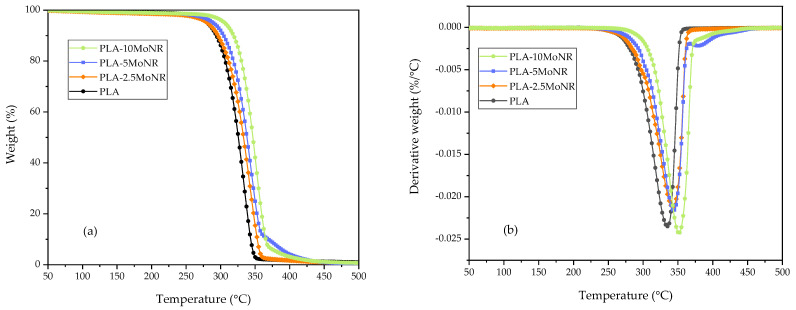
Thermal decomposition of neat PLA and PLA-MoNR at various MoNR contents. (**a**) TGA thermograms and (**b**) DTG thermograms.

**Figure 11 polymers-16-00812-f011:**
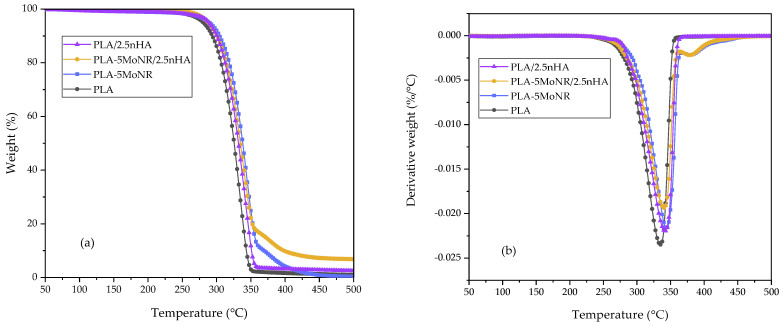
Thermal decomposition of neat PLA, PLA-5MoNR, and PLA/nHA composites with/without MoNR. (**a**) TGA thermograms and (**b**) DTG thermograms.

**Table 1 polymers-16-00812-t001:** The compositions of neat PLA, PLA-MoNR at various MoNR contents, and PLA/nHA composites with/without MoNR.

Designation	PLA (phr ^1^)	MoNR (phr ^1^)	nHA (phr ^1^)
PLA	100	-	-
PLA-2.5MoNR	100	2.5	-
PLA-5MoNR	100	5	-
PLA-10MoNR	100	10	-
PLA-5MoNR/2.5nHA	100	5	2.5
PLA/2.5nHA	100	-	2.5

^1^ phr refers to parts per hundred resins.

**Table 2 polymers-16-00812-t002:** Elemental composition of nHA powder.

Elements	Atomic %
O	63.95
P	13.24
Ca	22.81

**Table 3 polymers-16-00812-t003:** Mechanical properties of neat PLA, PLA-MoNR at various MoNR contents, and PLA/nHA composites with/without MoNR.

Designation	Young’s Modulus (MPa)	Tensile Strength (MPa)	Elongation at Break (%)	Impact Strength (kJ/m^2^)
PLA	599 ± 70	72.5 ± 4.9	14.8 ± 3.9	2.7 ± 0.1
PLA-2.5MoNR	504 ± 85	68.7 ± 12.0	18.0 ± 1.0	2.7 ± 0.2
PLA-5MoNR	455 ± 94	48.4 ± 4.8	75.5 ± 13.0	3.1 ± 0.4
PLA-10MoNR	454 ± 57	50.0 ± 2.5	39.1 ± 3.3	6.0 ± 0.5
PLA-5MoNR/2.5nHA	428 ± 43	46.6 ± 7.8	33.4 ± 4.1	3.3 ± 0.7
PLA/2.5nHA	527 ± 124	66.4 ± 4.9	16.1 ± 1.0	2.6 ± 0.2

**Table 4 polymers-16-00812-t004:** Thermal properties of neat PLA, PLA-MoNR at various MoNR contents, and PLA/nHA composites with/without MoNR.

Designation	First Heating			Second Heating		
	∆Hcc (Jg^−1^)	∆Hm (Jg^−1^)	Xc (%)	∆Hcc (Jg^−1^)	∆Hm (Jg^−1^)	Xc (%)
PLA	18.74	22.83	4.40	16.11	16.18	0.08
PLA-2.5MoNR	6.22	14.09	8.64	2.20	2.28	0.09
PLA-5MoNR	15.84	16.82	1.11	4.34	4.35	0.01
PLA-10MoNR	14.73	19.77	5.96	4.61	4.76	0.18
PLA-5MoNR/2.5nHA	13.41	14.73	1.53	3.24	3.47	0.27
PLA/2.5nHA	15.55	18.84	3.61	9.43	9.45	0.02

**Table 5 polymers-16-00812-t005:** Thermal properties of neat PLA, PLA-MoNR at various MoNR contents, and PLA/nHA composites with/without MoNR obtained from second heating scan.

Designation	Tg *_PLA_* (°C)	Tcc (°C)	Tm (°C)
PLA	61.33	126.33	152.67
PLA-2.5MoNR	61.33	133.00	154.33
PLA-5MoNR	61.33	133.00	154.33
PLA-10MoNR	60.33	133.00	154.33
PLA-5MoNR/2.5nHA	60.67	131.67	153.67
PLA/2.5nHA	62.33	131.67	156.00

**Table 6 polymers-16-00812-t006:** Thermal stability in terms of $T_{Onset}$ and ${\;T}_{Max}$ of neat PLA, PLA-MoNR at various MoNR contents, and PLA/nHA composites with/without MoNR.

Designation	$T_{Onset}$ (°C)	$T_{Max}$ (°C)
PLA	302.82	334.83
PLA-2.5MoNR	311.23	342.67
PLA-5MoNR	314.64	343.83
PLA-10MoNR	326.71	351.50
PLA-5MoNR/2.5nHA	304.92	340.00
PLA/2.5nHA	306.48	341.67

## Data Availability

No new data were created or analyzed in this study. Data sharing is not applicable to this article.

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
