# Peer review of "Effect of Modified Natural Rubber on the Mechanical and Thermal Properties of Poly(Lactic Acid) and Its Composites with Nanoparticles from Biowaste"

_polymers, 2024, doi:10.3390/polym16060812_

Round 1

Reviewer 1 Report

Comments and Suggestions for Authors

The manuscript id deal with the effect of natural rubber grafted copolymer on the mechanical and thermal properties of PLA and PLA composite. in spite the manuscript deals with preparation of eco-friendly material with sustainable resources, however the manuscript presentation should be improved to be suitable for final publication as follows 

1- English languauge should be revised all over the manuscript.

2- the abbreviation should be avoided as possible all over the manuscript as it make a confusion for the reader or add abbreviation list .

3- the abstract section should be rewritten to be more clear and identify the main idea and the target of the manuscript.

4- the manuscript target is to improve the mechanical properties of the PLA through either the addition  of the modified natural rubber or the naturally extracted Hydroxaptite, so, all the material characterization sections should be included onto one chracterization section at the material & methods sections.

5- the morphology (SEM) results of all prepared materials either PLA or composite with NR or composite with HA should be compared onto one title at the result and discussion ection.

6- Also, XRD results for all prepared matrices should be compared regarding to the prepared free matrix and after composite preparation.

7- so, the results and discussion section should be re-investigated as comparable investigation for all studied characterization tools regarding to the free PLA compared with the composite PLA matrices.

Comments on the Quality of English Language

The manuscript should be re-written as comparable investigation as it is very confused for the reader due to the seperated discussion of each prepared matrix.

Reviewer 2 Report

Comments and Suggestions for Authors

The literary review is good and there is a good rationale for the work.

Flaw. The work is aimed at obtaining a cheap, durable polymer from natural raw materials, but the use of an expensive drying method does not correspond to the task. (Then, the precipitate was frozen at −80 °C and then lyophilized by freeze dryer (Alpha 2−4 LSC, Chirist, Germany) for 1 day. The final product become the nanosized HA (nHA).)

  A routine scientific study has been carried out and the results are described. Outstanding results were not obtained. General, average study. May be published with a low rating

Reviewer 3 Report

Comments and Suggestions for Authors

Presented work is related to the composites obtained using PLA, chemically modified NR and hydroxyapatite nanoparticles obtained from fish scales. The work is well designed and interesting, but should be completed in some points:

- Section 2.2 should be completed by the yield of the process (mass of hydroxyapatite / mass of fish scales)

- Table 1: Please, correct the sample code of 5th sample

- Table 3: Please, round the results to two significant digits (starting from standard deviation

- Static tensile test: Please, present the exemplary stress-strain curve for each material (in one plot)

Round 2

Reviewer 1 Report

Comments and Suggestions for Authors

The manuscript was enhanced and modified. So its accepted for publication in polymers.